# TIME SERIES FORECASTING: EMPOWERING EXOGENOUS DATA WITH SHAPE MORPHING

## ABSTRACT

Time series forecasting often relies on patterns extracted from historical target dynamics, yet exogenous variables can provide valuable additional signal. Importantly, such variables are typically informative only in specific intervals and irrelevant elsewhere. We refer to this phenomenon as temporal saliency of exogenous variables, i.e., the time-varying relevance of external inputs for predicting the target series. In this paper, we tackle the "forecasting with exogenous variables" problem, where the model receives multiple input channels but predicts only one target variable. Recent studies have shown that channel-dependent Transformer architectures might be outperformed by simple channel-independent linear models, suggesting that current cross-attention mechanisms suffer to fully profit from exogenous information. To address this, we propose a morphing framework that adaptively reshapes exogenous time series before forecasting. For each channel and time step, a morphing function computes a ratio from the local relationship between the exogenous input and the target series and amplifies useful intervals accordingly. We instantiate morphing functions with interpretable information-theoretic metrics such as correlation, covariance, entropy, and mutual information, and evaluate them in ablation studies for long-horizon forecasting and state-of-the-art Transformer-based architectures. Results show that morphing is capable of yielding significant improvements in certain dataset–model combinations. These findings highlight morphing as a simple yet effective way to enhance the utility of exogenous information and close part of the performance gap between linear and Transformer-based forecasting methods.

## 1 INTRODUCTION

Time-series forecast plays a key role in modern decision-support approaches. Traditionally, forecasts are based on statistical information from historical data. Yet, the forecast accuracy depends mainly on the quality of the model, which describes the target variable. Thus, enhanced approaches additionally include external information, derived from variables with relation to the target variable. Typically, these variables influence the model of the target variable without mutual effect and are called exogenous dataDeistler & Scherrer (2022). In fact, accurate time-series forecasting requires a model that adequately describes the target variables.

The literature shows that exogenous variables are capable of enhancing model accuracy and thus contributing to more precise forecasts. Numerous studies in different domains conclude the benefit of exogenous variables as additional inputs in time-series models as shown by several reviews such as in Christen et al. (2020) and de Luca Avila & De Bona (2020). Despite this, the consideration of additional variables in modelling entails careful selection of information. In fact, including additional information in modelling provides benefit only if the additional variables comprise information which describes some characteristics of the target variable; that means only when additional variables contain relevant information about the behaviour of the target variable. Consequently, the model must be capable of recognising and extracting the extra information from exogenous series in order to benefit and enhance the model's accuracy.

Recent leveraging Artificial Intelligence (AI) methods originating from language models, literally transformer models, also present remarkable results when applied for time series forecasting. However, transformer models lack the ability of reasonably identifying and integrating peculiarities from

exogenous data with significant influence on the target variable. In particular, transformer models are permutation-invariant that contradicts the time continuity, a key property of time series. Consequently, these models hardly benefit from the additional information provided by exogenous variables. This effect is presented and discussed in numerous studies such as in Duong-Trung et al. (2024); Nie et al. (2023); Lu et al. (2024); Han et al. (2023), to name a few. As a result, several approaches focus particularly on saliency detection such as in Lim et al. (2020); Duong-Trung et al. (2024); Lu et al. (2024).

For mitigating blindness of transformer models in terms of temporal peculiarities, we developed a new approach that provides information about temporal peculiarities of exogenous series to the transformer model. In fact, the approach identifies temporal saliencies of exogenous series in a preprocessing step and feeds the morphed exogenous series to the input of the transformer model. In this way, the transformer model receives the information about influencing temporal peculiarities from the input and is able to dedicate the focus on the behaviour of the target variable.

In this paper we present a new morphing approach that emphasizes exogenous information temporally based on its influence on the target variable. The paper discusses the concept, limitations and potential of the new approach towards further research. In an extensive ablation test, we evaluate the efficiency of the proposed approach for long-term time series forecast on seven data sets comprising several exogenous variables, five transformer models and five methods for saliency detection. Results indicate additional supportive behaviour with morphed exogenous variables in general. Notable improvements were achieved in the Crossformer models, but also in the Autoformer and iTransformer models, which significantly benefit from the proposed approach to increase prediction accuracy.

The remainder of this paper is structured as follows: Section 2 addresses the topic of saliency detection in time series and the use of attention weights for exogenous data in transformer models. Subsequently, in Section 3 we present the main concept of the proposed morphing approach before evaluating and reasoning about the potential of the new approach with several experiments in Section 4. Finally, Section 5 summarizes the findings of the experiments for further research.

## 2 ATTENTION WEIGHTS IN EXOGENOUS DATA

Modelling a particular time series with historical and exogenous data implies accurate identification of peculiarities in all input of the considered time series. Exogenous variables effect the model but are not affected by it itself, as defined in Deistler & Scherrer (2022). Although these variables can contribute to more accurate modelling, valuable information from these variables need to be carefully identified and integrated into the modelling process. Data from exogenous variables can comprise relevant information that describes certain characteristics of a response variable. Yet, the full series is typically only partially related to the target variable. This means that only certain time intervals significantly influence the target series. Others rarely affect the target variable or have no relationship with it at all.

In the literature, several works in different realms demonstrate the potential of exogenous data for increasing forecast accuracy. Studies on time series forecast with a focus on financial data de Luca Avila & De Bona (2020), groundwater level Hoque et al. (2024), nitrogen dioxide ($NO_2$) González-Enrique et al. (2021) or power load forecasting Lee & Cho (2022); Christen et al. (2020), to name a few, all conclude the improvement in forecast accuracy by incorporating exogenous variables. In contrast, studies discuss the issue of appropriate variable compositions used for time series modelling Álvarez Chaves et al. (2024); Sowinski (2021). In Bento et al. (2022) Bento et. al even in numerical tests a disconnect between the best input data combination and the common Pearson correlation analysis.

As a consequence, using exogenous data as predictors coerces models to identify and incorporate peculiarities from exogenous series proportional to their relevance. In fact, the characteristics of a target time series result from a linear combination of exogenous data influences. Based on this fact, recent research on forecasting with exogenous variables pays more attention to the identification and relevant dependent consideration of temporal peculiarities, as discussed in Duong-Trung et al. (2024). The first approaches date back to the 1970s, when Box and Jenkins introduced an extension of the Autoregressive Integrated Moving Average (ARIMA) statistical model for exogenous data,

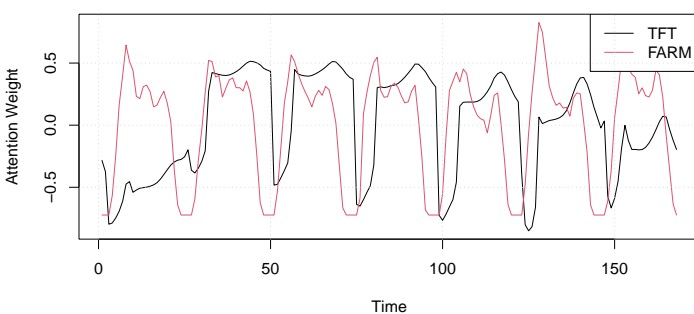

Figure 1: Comparison of attention weights for temporal salience discovered from the Temporal Fusion Transformer (TFT) model and the statistical method used in the Forward Angular Relevance Measure (FARM).

ARIMAX Kendall et al. (1971). Modern approaches have successfully adopted transformer models with self-attention mechanisms for time series forecasting problems. The attention computation allows direct pairwise comparison with any unusual occurrence and can inherently model temporal dynamics Duong-Trung et al. (2024). Aiming for explainability and increased forecast accuracy, saliency maps have been widely used to highlight important peculiarities in time series used in modeling Lim et al. (2020); Ismail et al. (2021); Pan et al. (2021). Duong-Trung et al. proposed a new method, inspired by saliency detection theory in images and video recognition, to weigh the proper attention to possible emerging temporal patterns Duong-Trung et al. (2024). The authors discuss the need for a technique to automatically encode saliency-related temporal patterns by connecting them to the appropriate attention heads.

In an experiment with pedestrian counts in touristic hotspots in a city, we compared the attention weights provided by the Temporal Fusion Transformer (TFT) model Lim et al. (2020) with purely statistically identified saliencies. With this experiment, we aim to reduce the complexity of input variables arising from individual dependencies. The comparison applied the Forward Angular Relevance Measure (FARM) algorithm Auth1 et al. (2023) for statistical saliency detection which results in a series of correlation coefficients from a rolling window. Figure 1 presents the attention weights identified by algorithms TFT and FARM for the hourly samples over one week. Each attention weight is averaged with the weights from the same point in time over multiple weeks. In particular, the comparison exposes remarkably similar patterns with a small but constant time shift in the extracted attention weights. Consequently, we hypothesise that statistical methods allow for comparable saliency detection. This enables decoupling saliency detection from modelling and applying transformer models solely on modelling the behaviour of the target variable based on features provided by exogenous variables. Based on the results of recent studies with comparable approaches Pan et al. (2021); Lu et al. (2024), we expect a more efficient use of exogenous information to increase the accuracy of time series forecasts.

## 3 MORPHING EXOGENOUS INPUT DATA

Modelling time-series variables involves describing the characteristics of the target variable. This also includes characterising an action under specific external influences. More precisely, models can benefit from exogenous variables and yield greater accuracy by including them in the input variables, as discussed in the previous sections. Transformer models' main strength is their multi-head self-attention mechanism, which can automatically learn the connections between elements in a sequence. This property makes transformer models ideal for sequential modelling tasks Nie et al. (2023). They show a remarkable capability of extracting semantic among elements in a long sequence such as words in texts. Yet, transformer models struggle with identifying interactions between multiple input variables, due to the permutation invariance property. This implies that transformer models lack the ability to maintain features in the strict order in which they appear

in the input time series. In contrast, these models have the remarkable ability to extract permuted semantic correlation between elements in a long sequence, as appears in text or 2D fragments in images Zeng et al. (2023). However, time series typically lack of semantics in the numerical data itself, and modelling instead focusses on temporal changes among a continuous set of points Zeng et al. (2023). In fact, transformer models are not capable of learning channel dependency. Several studies in the literature show that univariate models, which ignore inter-series relationships, often outperform their multivariate counterparts Lu et al. (2024). Thus, Channel Independent (CI) models frequently outperform Channel Dependent (CD) models by a significant margin, as shown by a comprehensive empirical and theoretical analysis in Han et al. (2023). Consequently, following the findings and proposals for statistical data preprocessing in Rana & Odum (2025), we propose a new approach for CD modelling with statistical input preprocessing.

Based on the hypothesis derived in Section 2, morphing exogenous series with respect to their influence on the target variable might enable their usability for transformer models. The proposed approach builds on a preceding statistical analysis to identify the temporal influence of a single exogenous series ($x_t$) on the target variable ($y_t$). In this analysis, a statistical function $S$ iteratively identifies in a small window ($w$) the similarity of a small section between the target and exogenous series. In doing so, according to Equation 1, the statistics yield a series of positive numbers that indicate the required level of attention ($r_t^{(k)}$) for each data point ($k$) to include exogenous series with temporal attention in the modelling.

$$r_t^{(k)} = S\big(x_{t-w+1:t}^{(k)}, y_{t-w+1:t}\big) \tag{1}$$

Subsequently, the proposed approach applies the temporal attention factors on the original exogenous time series by morphing the amplitude values accordingly. In fact, the morphing ($\mathcal{M}$) amplifies or decreases each data value of the exogenous time series according to the statistically identified temporal influence (similarity) of the exogenous series, as expressed in Equation 2. The detailed formal definition of the morphing framework is given in the appendix in Section D.

$$\mathcal{M}_{S,w}\big(x^{(k)}, y\big)_t = r_t^{(k)} \cdot x_t^{(k)} \tag{2}$$

Finally, the morphed exogenous series allows transformer models to integrate influence-adjusted exogenous information in modelling of the target series. As a consequence, transformer models can focus on the connections between peculiarities in the target time series without the need of identifying the influence from exogenous variables.

To illustrate the effect of the proposed morphing approach, we constructed a synthetic toy example with a single exogenous series and a univariate target series. The target $y_t$ combines autoregressive and seasonal components, while the exogenous feature $x_t$ only influences $y_t$ in specific time intervals (both positively and negatively), and remains irrelevant elsewhere. These relevant regimes are highlighted in the background of Figure 2(a). We then computed a *lag-aware rolling correlation* between $x_t$ and $y_t$, smoothed it with an exponential moving average, and mapped it into a *morph ratio* $r_t \in [0.3, 1.7]$. As shown in Figure 2(b), the morph ratio rises above 1 during intervals when $x_t$ is predictive of $y_t$, and drops to 1 or less when the exogenous feature is irrelevant or misleading. Multiplying the exogenous series by this ratio produces a *morphed series* $\tilde{x}_t = r_t x_t$, shown in Figure 2(c), where the contribution of $x_t$ is amplified in useful regimes and attenuated in irrelevant ones. Finally, we trained a simple Ridge regression forecaster using lagged values of both the target and the exogenous inputs. Figure 2(d) compares forecasts obtained with the original exogenous series and with the morphed series: while both models capture the general dynamics of $y_t$, the morphed version achieves visibly closer alignment during the highlighted intervals. Quantitatively, the mean squared error on the held-out test set decreased by approximately 6%, demonstrating how morphing can help even a simple linear model better exploit exogenous information when it is truly informative, while suppressing noise in irrelevant periods.

To the best of our knowledge, morphing exogenous variables based on statistically identified influences of exogenous characteristics on the target variable has never been discussed in the literature to date. Yet, recent publications present an improvement in forecast accuracy and a reduction in model complexity with statistical preprocessing of input time series. Such as in Rana & Odum (2025) where Rana and Odum improved channel-independent transformer models with statistical prepro-

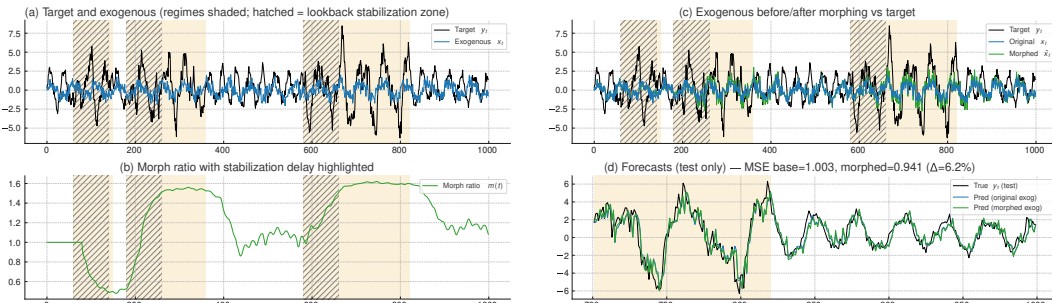

Figure 2: Synthetic toy example illustrating the proposed morphing approach. (a) Target series $y_t$ and an exogenous series $x_t$, with shaded regions indicating time intervals where $x_t$ truly influences $y_t$. (b) Morph ratio $r(t)$ computed from lag-aware rolling correlation, which rises during relevant regimes and falls elsewhere. (c) Morphed exogenous series $\tilde{x}_t = r_t \, x_t$, where the influence of $x_t$ is amplified when useful and attenuated when irrelevant. (d) Forecast comparison on the test set using a simple Ridge regression forecaster. Incorporating the morphed exogenous improves alignment with $y_t$ and reduces the mean squared error by about 6% compared to using the original exogenous series.

cessing or the CATS framework that yield improvements with the construction of auxiliary time series Lu et al. (2024), respectively. Yet, our experiments presented and discussed in the subsequent Section 4 yield comparable results and thus support the theory presented in the recent literature. A comparison between the proposed morphing approach and the CATS framework is given in the appendix in Section E.

## 4 EXPERIMENTS

The proposed morphing approach for exogenous data, introduced in previous Section 3, aims to reduce the complexity of the modelling task for AI approaches. Indeed, a preliminary statistical analysis of the interactions between input variables should allow modelling to focus more on identifying continuous characteristics than on the interaction of variables. Statistical preprocessing results in exogenous data series with modified amplitude. With modified exogenous series, the morphing approach should support AI in modelling the ecosystem of target series and improve the accuracy of the forecasts. The prefaced analysis should especially benefit transformer models that struggle with identifying variable interactions, mainly resulting from the permutation-invariance property.

In this experiment, we conducted an extensive ablation test to evaluate the effect of morphing exogenous input data based on temporal saliencies. The experiment relies on seven data sets, widely used in literature for contrasting forecast performance, and varies in different parameters having potential effect on the morphing process and the forecast accuracy. In addition, to evaluate the benefit of morphing input data on different AI characteristics, the ablation test includes seven transformer models, which gained popularity in recent research on the topic of time series forecasting.

In particular, the experiment applies seven well-established data sets used in the benchmark of the comprehensive deep time series review Wang et al. (2024a) for long-term time series forecasting task. The data include: 1) an Electricity Consumption Load (ECL) data set with an hourly recording comprising 26'304 observations of a power consumption (target) and 320 exogenous variables; 2-3) two Electricity Transformer Temperature (ETT) data sets with hourly recordings (ETTh1 and ETTh2) of the transformer oil temperature in 17'420 observations and with six related exogenous variables; 4-5) two ETT data sets of the same transformer oil temperatures and exogenous variables with 15 minutes recording intervals (ETTm1 and ETTm2) resulting in 69'680 observations; 6) a traffic load data set with hourly recordings of 861 exogenous variables and one target variable in 17'544 observations; and 7) a weather data set, consisting of 20 exogenous variables and one target variable, with a 10 minutes recording interval over 52'696 observations. All data sets have the target variable named with OT. A detailed overview of the data sets used in the experiment is given in the appendix in Section A.

In the experiment, we used the historical values of the target and all exogenous variables as input to forecast the next value of the target variable. Specifically, we trained multivariate models with univariate output. The literature calls this type of approach as *exogenous model* to differentiate it from multivariate approaches with multiple input and multiple output variables. For long term n-multipoint forecasts, we calculated n single-point forecasts in an iteration and extended the input data of the target variable with the forecast of the previous iteration ($fc_{[n-1]}$) for the calculation of the next forecast ($fc_{[n]}$). Yet, the exogenous data always comprise the original information. This procedure is coherent with the process for exogenous time series forecasting and complies with related literature such as Wang et al. (2024b).

The forecast intervals include 96, 192, 336 and 720 data values for each data set. In doing so, we applied the entire ablation test across the full set of exogenous variables to each of the different forecast ranges. In addition, the experiment also applies a set of different saliency detection algorithms and varies the size of the sliding window for temporal saliency detection throughout the length $l \in \{501, 751, 1001, 1251, 1501\}$. Thereby, the point of interest is set on the last data point in the sliding window in order to retain it applicable in real case scenarios with continuous data processing and forecasting. In fact, the temporal saliency attention weights ($r_t^{(k)}$) are calculated according to Equations 1.

The applied statistics ($\mathcal{S}$) used for temporal saliency detection involve the following information measures: FARM, mutual information (MI), covariance, correlation and entropy. Finally, the experiment evaluates the forecast performance with the three error measures Mean Absolute Error (MAE), Mean Absolute Percentage Error (MAPE), and Mean Squared Error (MSE), which are widely used in the literature for contrasting time series forecasting methods. For straight forward and qualitative evaluation of the shape morphing approach, we applied the same hyperparameters as used in the literature according to the original paper of each model. [1] For reproducibility, details about the experiment setup are given in the Appendix in Section A.

The following sections discuss the results of the experiments with respect to the effects of the shape morphing approach in Subsection 4.1 and the sliding window length in Section 4.2. We also applied significance tests to support our findings. The results are given in the appendix in Section C.

## 4.1 MAIN RESULTS

The shape morphing approach applies relevance factors of the identified temporal saliencies in exogenous time series on its amplitude value. Including amplitude-shaped exogenous time series as additional inputs in exogenous forecast methods, the morphing approach aims to support training with additional information. In doing so, the proposed approach separates saliency detection in exogenous series, thus the analysis of interactions, into a statistical preprocessing. As a result, modelling can focus on learning the behaviour of the target time series based on peculiarities in the exogenous and history data.

Table 1 presents a summary of the improvements obtained in time series forecasting by including exogenous information with morphed amplitude values. For all data sets included in the experiment, the table shows forecast accuracies for 96, 192, 336, and 720 forecasted data points obtained with morphed exogenous data across five transformer models that recently gained popularity in time series forecasting. In this comparison, the forecast accuracy is expressed as MAE along with a relative number indicating the gain compared to a forecast without exogenous information. That means that the given forecast accuracy presents the best result of the performed ablation test obtained with the optimal configuration of exogenous variables. Missing accuracy values are the result of an overload of the experiment environment. Indeed, the large number of exogenous variables in the ECL and TrafficL data sets exhausted the system used for the experiments and resulted non-comparable accuracy rates.

Comparing forecast accuracy rates across all experiments reveals improvements in 86 cases out of 117, which is approximately 73% of all experiments. In particular, morphing the exogenous series with attention weights of detected saliencies yields significant improvements of $\geq 80\%$ of the forecasting experiments with the data sets ETTh1, ETTh2, ETTm2, and TrafficL. To put it differently, from the perspective of applied AI methods the morphing approach results in remarkable improve-

---

[1]Parameters will be published along with the code after acceptance.

Table 1: Time series forecast accuracy (MSE) of the experiment across all applied data sets, the various forecast horizons ($n$ time steps), and modelling (AI). Next to the accuracy measure, the relative values indicate the improvement in the forecast accuracy. Notable are the benefits for the Crossformer models ($\phi + 31.9\%$), across nearly all datasets and forecast horizons, as well as for certain Autoformer and iTransformer models. The vast number of exogenous variables in the ECL and TrafficL data set exhausted the computational power of the machine used for the experiments and yield non-comparable accuracy results.

| Data Set | Horizon | Autoformer | | Crossformer | | PatchTST | | TimeXer | | iTransformer | |
|---|---|---|---|---|---|---|---|---|---|---|---|
| ECL | 96 | 0.3244 | 12.5%↑ | - | | 0.3129 | 0.0% | 0.2659 | 1.1%↑ | 0.2730 | 2.1%↑ |
| | 192 | 0.4565 | 3.2%↑ | - | | - | | 0.3160 | -0.7%↓ | 0.3123 | 1.4%↑ |
| | 336 | 0.4069 | 10.1%↑ | - | | - | | 0.3628 | 1.9%↑ | 0.3608 | 3.9%↑ |
| | 720 | 0.5175 | -15.1%↓ | - | | - | | 0.3777 | -3.8%↓ | 0.4321 | 5.7%↑ |
| ETTh1 | 96 | 0.0807 | 30.7%↑ | 0.1689 | 54.5%↑ | 0.0567 | 0.1%↑ | 0.0552 | 2.4%↑ | 0.0570 | 2.8%↑ |
| | 192 | 0.0866 | 6.8%↑ | 0.1671 | 59.4%↑ | 0.0744 | -0.0%↓ | 0.0780 | -6.4%↓ | 0.0703 | 3.0%↑ |
| | 336 | 0.1180 | 0.2%↑ | 0.2517 | 14.6%↑ | 0.0883 | 0.1%↑ | 0.0858 | -0.5%↓ | 0.0797 | 5.1%↑ |
| | 720 | 0.1177 | 8.9%↑ | 0.2439 | 26.4%↑ | 0.0902 | -0.0%↓ | 0.0871 | 7.6%↑ | 0.0963 | 4.9%↑ |
| ETTh2 | 96 | 0.1474 | 25.1%↑ | 0.2223 | 58.2%↑ | 0.1358 | 0.4%↑ | 0.1315 | 8.9%↑ | 0.1320 | 3.7%↑ |
| | 192 | 0.2160 | 2.2%↑ | 0.4755 | 54.1%↑ | 0.1924 | 1.8%↑ | 0.1812 | 5.4%↑ | 0.1830 | 5.6%↑ |
| | 336 | 0.2711 | -1.0%↓ | 0.7100 | 50.0%↑ | 0.2276 | 2.1%↑ | 0.2336 | 3.1%↑ | 0.2153 | -0.2%↓ |
| | 720 | 0.2706 | -0.7%↓ | 1.8190 | -2.0%↓ | 0.2450 | 5.0%↑ | 0.2193 | 4.6%↑ | 0.2589 | 0.8%↑ |
| ETTm1 | 96 | 0.0546 | 6.7%↑ | 0.0543 | 31.8%↑ | 0.0292 | 0.6%↑ | 0.0282 | 0.8%↑ | 0.0299 | 1.5%↑ |
| | 192 | 0.0717 | -0.0%↓ | 0.2107 | 28.8%↑ | 0.0457 | -4.1%↓ | 0.0462 | -2.6%↓ | 0.0445 | 0.2%↑ |
| | 336 | 0.0855 | 0.9%↑ | 0.4364 | 13.8%↑ | 0.0588 | 1.3%↑ | 0.0596 | -1.0%↓ | 0.0619 | 0.6%↑ |
| | 720 | 0.1028 | -3.4%↓ | 0.8312 | 11.7%↑ | 0.0855 | -5.9%↓ | 0.0816 | -1.2%↓ | 0.0803 | 0.2%↑ |
| ETTm2 | 96 | 0.1128 | 8.1%↑ | 0.0977 | 56.0%↑ | 0.0637 | 1.1%↑ | 0.0660 | 2.5%↑ | 0.0681 | 10.1%↑ |
| | 192 | 0.1512 | 3.1%↑ | 0.1333 | 76.6%↑ | 0.1013 | 1.2%↑ | 0.1017 | 1.9%↑ | 0.1047 | 12.9%↑ |
| | 336 | 0.1485 | 2.5%↑ | 0.3864 | 54.0%↑ | 0.1324 | 0.5%↑ | 0.1340 | 1.2%↑ | 0.1408 | 10.5%↑ |
| | 720 | 0.1879 | 9.7%↑ | 1.0020 | 32.0%↑ | 0.1876 | 0.6%↑ | 0.1899 | -0.7%↓ | 0.1836 | 4.8%↑ |
| TrafficL | 96 | 0.2349 | 6.7%↑ | - | | - | | 0.1640 | 2.7%↑ | 0.1477 | 5.6%↑ |
| | 192 | - | | - | | - | | - | | 0.1550 | 0.3%↑ |
| Weather | 96 | 0.0051 | 34.9%↑ | 0.0014 | 76.6%↑ | 0.0012 | 20.1%↑ | 0.0012 | 2.8%↑ | 0.0012 | 9.1%↑ |
| | 192 | 0.0161 | -116.7%↓ | 0.0039 | -5.3%↓ | 0.0017 | 1.5%↑ | 0.0015 | 1.6%↑ | 0.0015 | 5.2%↑ |
| | 336 | 0.0112 | -101.4%↓ | 0.0021 | -6.7%↓ | 0.0019 | -1.5%↓ | 0.0016 | 1.6%↑ | 0.0017 | -3.1%↓ |
| | 720 | 0.0048 | - 5.7%↓ | 0.0041 | -46.7%↓ | 0.0022 | -0.9%↓ | 0.0021 | 0.9%↑ | 0.0022 | -0.2%↓ |

ments for the Crossformer of $+31.9\%$ in average and for certain forecasts with the Autoromer and the iTransformer models, respectively.

Furthermore, the results show that especially short forecast horizons ($n = 96$ samples) benefit from morphing the exogenous series in a preprocessing step. Without any performance loss across all experiments, the shortest forecast horizon clearly profits most from the morphing approach, followed by the intermediate forecast horizons with 192 and 336 data points, where seven forecasts obtained reduced accuracy, and the large horizon with 720 forecasted data points where 13 out of 28 experiments result reduced forecast accuracy. Nevertheless, more than $50\%$ still benefits from the proposed morphing approach.

## 4.2 TEMPORAL SALIENCY DETECTION

Morphing exogenous series based on the relevance of temporal saliencies requires information about peculiarities that influence the target time series. This involves identifying temporal peculiarities in the exogenous series that are common to the target time series and estimating their influence on the target time series. The ablation study inspects all exogenous series of a data set separately in a direct comparison with the target variable. In doing so, we applied four different methods for identifying temporal saliencies: FARM, MI, covariance, correlation and entropy. The identification of temporal saliencies moves a sliding window over compared time series that sequentially estimates the proportion of shared information within the time frame. Thereby, the resulting proportion of shared information is a measure for point-wise attention weight of temporal saliency and is mapped to the last data point of the sliding window. Thus the method complies with the requirements for continuous application on streaming time series.

Of all the results of the experiments, Table 2 summarises the five best combinations of the modelling and saliency detection methods in all data sets. The selection of the five best combination relies on the MAE performance measure. In combination with the proposed morphing approach, the Table 2 clearly presents a dependency between the performance quality of the model and the applied data set. In five out of seven data sets, there is one particular model, which yields the most accurate forecasting results. In contrast, the saliency detection method is more heterogeneous without an outperforming method. This holds for positive as well as inverted attention weights which are marked with the prefix *i*. Nevertheless, all data sets benefit from the proposed morphing approach with the statistical identification of temporal saliencies in preprocessing. In fact, no modelling approach without morphed input data was able to compete among the top five models with the lowest forecast error rates.

Table 2: Summary of the five best performing model and saliency detection combinations along all data sets used in the experiments. Remarkably, most data sets have one particular model that ranks within the five most accurate forecast results. Additionally, all leading models benefit from the mophing approach and forecasts with non-preprocessed input data are unable to compete with the best statistical saliency detection methods.

| Dataset | Model | Saliency Detection | MAE ↑ | MSE | MAPE |
|---------|-------|--------------------|-------|-----|------|
| ECL | TimeXer | ipfarm | 0.370496 | 0.265915 | 1.847979 |
| ECL | TimeXer | pmutual_info | 0.372232 | 0.266670 | 1.840901 |
| ECL | TimeXer | pfarm | 0.370433 | 0.266690 | 1.810957 |
| ECL | TimeXer | ipfarm | 0.371691 | 0.266844 | 1.787365 |
| ECL | TimeXer | prollcov | 0.370650 | 0.267423 | 1.819801 |
| ETTh1 | TimeXer | pmutual_info | 0.179195 | 0.055289 | 0.142015 |
| ETTh1 | TimeXer | pmutual_info | 0.181302 | 0.056045 | 0.144518 |
| ETTh1 | TimeXer | pmutual_info | 0.181624 | 0.056093 | 0.145285 |
| ETTh1 | TimeXer | prollcov | 0.181525 | 0.056142 | 0.144778 |
| ETTh1 | TimeXer | prollcov | 0.181708 | 0.056147 | 0.145127 |
| ETTh2 | TimeXer | prollcov | 0.278411 | 0.131487 | 0.689359 |
| ETTh2 | iTransformer | prollcorr | 0.280502 | 0.131999 | 0.692239 |
| ETTh2 | TimeXer | prollcov | 0.279467 | 0.132492 | 0.691322 |
| ETTh2 | iTransformer | pmutual_info | 0.281887 | 0.132525 | 0.695883 |
| ETTh2 | TimeXer | pmutual_info | 0.281816 | 0.132750 | 0.685258 |
| ETTm1 | TimeXer | iprollcorr | 0.125775 | 0.028265 | 0.103218 |
| ETTm1 | TimeXer | ipfarm | 0.125760 | 0.028287 | 0.103338 |
| ETTm1 | TimeXer | pentropy | 0.125726 | 0.028322 | 0.103423 |
| ETTm1 | TimeXer | ipmutual_info | 0.125734 | 0.028323 | 0.103351 |
| ETTm1 | TimeXer | ipmutual_info | 0.125964 | 0.028366 | 0.103381 |
| ETTm2 | PatchTST | prollcorr | 0.183264 | 0.063732 | 0.466763 |
| ETTm2 | PatchTST | prollcov | 0.182854 | 0.063804 | 0.464072 |
| ETTm2 | PatchTST | pfarm | 0.182658 | 0.063815 | 0.466675 |
| ETTm2 | PatchTST | prollcov | 0.182900 | 0.063897 | 0.466614 |
| ETTm2 | PatchTST | pfarm | 0.183284 | 0.064036 | 0.465632 |
| TrafficL | iTransformer | iprollcorr | 0.230505 | 0.147665 | 0.807979 |
| TrafficL | iTransformer | iprollcorr | 0.233377 | 0.151606 | 0.814632 |
| TrafficL | iTransformer | iprollcorr | 0.234369 | 0.153041 | 0.807435 |
| TrafficL | iTransformer | prollcov | 0.234288 | 0.154417 | 0.803179 |
| TrafficL | iTransformer | prollcov | 0.234216 | 0.154460 | 0.807815 |
| Weather | iTransformer | ipmutual_info | 0.026184 | 0.001266 | 0.617928 |
| Weather | PatchTST | iprollcorr | 0.026344 | 0.001277 | 0.629863 |
| Weather | PatchTST | iprollcorr | 0.026596 | 0.001290 | 0.629982 |
| Weather | PatchTST | prollcov | 0.026507 | 0.001297 | 0.639845 |
| Weather | TimeXer | ipmutual_info | 0.026555 | 0.001297 | 0.635215 |

Identifying temporal saliencies entails dependency on the samples involved. Thus, applying a sliding window to restrict the samples considered in the evaluation of temporal interactions has a direct effect on the result. Measures that rely on average calculations tend to yield more balanced values as the number of samples increases. As a consequence, large moving windows better evaluate the influence of long lasting peculiarities, but obliterate strong temporary features with more intense short periods, compared to short moving windows. Table 3 summarises the moving window sizes of the best performing model - saliency detection setup. Among the considered moving windows sizes of 501, 751, 1001, 1251 and 1501 samples, the summary in the table shows roughly a balanced

distribution for the best forecast results with a slight advantage for small sizes. This might be explained by the blurring effect caused by large windows.

In the appendix, Table 5 in Section B provides an additional comprehensive summary of the accuracy of the forecast obtained for a fixed forecast length of 96 data points. This table presents comparable results with respect to the window size and also supports a slight advantage for small sizes.

Table 3: Moving window length of the best performing model - saliency detection setup across all applied data sets and AI models.

| model | Autoformer | Crossformer | PatchTST | TimeXer | iTransformer |
|---|---|---|---|---|---|
| ECL | 501 | - | 1251 | 501 | 1501 |
| ETTh1 | 501 | 1001 | 1001 | 751 | 751 |
| ETTh2 | 1251 | 501 | 1501 | 501 | 1251 |
| ETTm1 | 1251 | 1501 | 501 | 751 | 1501 |
| ETTm2 | 751 | 501 | 1001 | 1251 | 751 |
| TrafficL | 751 | - | - | 501 | 501 |
| Weather | 1001 | 1501 | 1501 | 501 | 1251 |

## 5 CONCLUSION

Forecasting time series requires accurate models of the target series. Such models can benefit from additional information provided by external variables, so-called exogenous variables, that support precise description of the target variable. Including these external variables into the modelling entails an exogenous modelling approach that identifies and learns interactions from exogenous variables on the target series. Most recent and promising AI modelling approaches originate from the domain of language processing (Large Language Model (LLM)) literally transformer models. However, these models struggle to learn the interactions necessary to adequately model the target series. To address this gap and enable time series forecasting to benefit from transformer models, we proposed and evaluated a morphing approach for exogenous data. The new approach first identifies and evaluates temporal saliencies in exogenous time series before adjusting the amplitude of the exogenous data based on the weights of the identified saliencies. These steps are applied separately to each exogenous variable before using them for modelling.

The evaluation consists of a large ablation test comprising seven data sets that contain several exogenous variables. All data sets are popular in the literature for time series forecasting. The results clearly demonstrate that transformer models, especially Autoformer, Crossformer and iTransformer models, benefit from the concept of morphing exogenous time series with saliency weights. Furthermore, the results show strong dependence between the transformer models and the applied data sets. However, the applied saliency detection and evaluation measures demonstrate balanced capabilities, with no clear trend towards any single exceptional method. The same holds for the applied window sizes used for the iterative detection of temporal saliency. We interpret the results as clear and strong evidence for a statistical preprocessing of exogenous time series that decouples saliency detection from modelling. The results of additional experiment that directly compare the proposed morphing approach with a DLinear neural network as baseline are presented in the appendix in Section F.

The proposed morphing approach combines two steps: first it identifies temporal saliencies in the exogenous series. Second, it adjusts the amplitude of the series according to the influence of each individual data point. Thus, our further research focusses on two aspects: 1. Refinement for temporal saliency detection and evaluation with statistics. This also includes the construction of the applied morphing ratio; in particular, the effect of a smoothed morphing ratio obtained by an applied smoothing function or by increasing the sliding window size. 2. Investigating the capabilities of how to best provide the attention weights in combination with exogenous information for transformer models.

Morphing is not universally better when used blindly (typical median effect $\approx 0\%$). However, when the morphing function and window size are chosen appropriately (i.e. tuned to an optimum) for a given dataset–model pair, it is capable of significantly boosting forecast accuracy. Thus, further research should rather focus on preprocessing of data instead of enhancing the complexity of new transformer models for detecting temporal saliencies.

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

# APPENDIX

## A  DATASETS, EXPERIMENT SETUP, AND REPRODUCIBILITY

### DATASETS

We conduct long-horizon forecasting experiments on **7 established real-world datasets** related to energy, climate, and traffic domains. A summary is given in Table 4.

- **ECL (Electricity Consumption Load)** Li et al. (2019): hourly electricity consumption data from 321 clients. The consumption of the last client is used as the target variable to be predicted, and the others serve as exogenous variables.
- **Weather** Zhou et al. (2021): meteorological data collected every 10 minutes at the Max Planck Biogeochemistry Institute in 2020. We use the Wet Bulb temperature as the target variable, with the remaining 20 indicators as exogenous variables.
- **ETT (ETTh1, ETTh2, ETTm1, ETTm2)** Zhou et al. (2021): power load and oil temperature datasets recorded in electricity transformers. ETTh1 and ETTh2 are recorded hourly, while ETTm1 and ETTm2 are recorded every 15 minutes. The oil temperature is the target variable, and the six power load features are exogenous.
- **Traffic** Wu et al. (2023): hourly road occupancy rates collected from 862 sensors on San Francisco Bay area freeways. The occupancy rate from the last sensor is used as the target variable, while the others are treated as exogenous variables.

Table 4: Dataset descriptions. Dataset sizes are reported as (Train, Validation, Test).

| Dataset | # Time series | Sampling Freq. | Dataset Size |
|---------|---------------|----------------|--------------|
| ECL     | 320           | 1 Hour         | (18317, 2633, 5261) |
| Weather | 20            | 10 Minutes     | (36792, 5271, 10540) |
| ETTh    | 6             | 1 Hour         | (8545, 2881, 2881) |
| ETTm    | 6             | 15 Minutes     | (34465, 11521, 11521) |
| Traffic | 861           | 1 Hour         | (12185, 1757, 3509) |

### EXPERIMENTATION AND REPRODUCIBILITY SETUP

All experiments were conducted using **Python 3.12.3** on **Linux Ubuntu 24.04.2 LTS**. The codebase is available at TSPindorama, a fork of Time-Series-Library. TSPindorama maintains compatibility with the original library while keeping a standard `requirements.txt` file with pinned package versions for reproducibility. In addition, it adds improved experiment configuration utilities and features that allow scaling to thousands of experiments without manual intervention, along with a large-scale results analysis tool. The implementation of the morphing preprocessing framework is provided separately at CommonDataSets.

All experiments ran on **AWS EC2 g6.xlarge** instances equipped with:

- 4 vCPUs and 16 GiB RAM for system resources,
- NVIDIA L4 GPU with ∼24 GiB VRAM as the accelerator.

The ablation study covers **7 datasets**, **5 models**, **6 morphing window lengths**, and **6 morphing functions**, each considered in both their **normal** and **inverted** forms, plus a *no-morphing* baseline at forecasting horizon 96. In total, this results in **2555 experiment runs**. From these, the best combinations of morphing function and window size for each dataset–model pair were re-evaluated at longer horizons (192, 336, 720). Each experiment was run only once per configuration, with a fixed random seed to ensure reproducibility. Importantly, all hyperparameters used for each model and dataset are **fully described in JSON configuration files**, which can be directly reused to replicate our experiments.

To further support reproducibility, we will release a **public Amazon Machine Image (AMI)** upon paper acceptance, along with:

- the complete source code,

- the original datasets,

- the shape-morphed datasets generated during preprocessing,

- all JSON configuration files describing the hyperparameters.

Our experiments also partially reproduce results from the TimeXer paper Zhou et al. (2021), though exact replication was not possible due to environment differences. Crucially, under this consistent setup, we observe that incorporating the morphing framework reliably improves forecasting performance.

## B    EFFECTS OF MORPHING ON ACTUAL FORECASTS

The following selected results contain the best morphing functions and window lengths for each dataset-model. Table5 provides error values and improvements obtained with specific combinations of morphing.

Table 5: Overall best MSE accuracy morphing function and window length combinations across all different dataset-model pairs for prediction length 96.

| Dataset | Model | Morphing function | Pred Len | Improvement | Window | MSE (morphed) | MSE (identity) |
|---|---|---|---|---|---|---|---|
| ECL | Autoformer | Inverted FARM | 96 | 12.513328 | 501 | 0.324377 | 0.370773 |
| ECL | PatchTST | Inverted smooth random walk | 96 | 2.173077 | 501 | 0.306052 | 0.312850 |
| ECL | TimeXer | Inverted FARM | 96 | 1.080630 | 501 | 0.265915 | 0.268820 |
| ECL | iTransformer | Mutual Information | 96 | 2.087391 | 1501 | 0.272955 | 0.278774 |
| ETTh1 | Autoformer | Mutual information | 96 | 30.653632 | 501 | 0.080700 | 0.116373 |
| ETTh1 | Crossformer | Mutual information | 96 | 54.481525 | 1001 | 0.168931 | 0.371127 |
| ETTh1 | PatchTST | Inverted smooth random walk | 96 | 1.581203 | 751 | 0.055874 | 0.056771 |
| ETTh1 | TimeXer | Mutual Information | 96 | 2.382867 | 751 | 0.055290 | 0.056639 |
| ETTh1 | iTransformer | Mutual Information | 96 | 2.812070 | 751 | 0.057038 | 0.058688 |
| ETTh2 | Autoformer | Inverted rolling covariance | 96 | 25.063986 | 1251 | 0.147415 | 0.196721 |
| ETTh2 | Crossformer | Random walk | 96 | 66.491784 | 501 | 0.178256 | 0.531978 |
| ETTh2 | PatchTST | Mutual information | 96 | 0.369914 | 1501 | 0.135849 | 0.136353 |
| ETTh2 | TimeXer | Rolling covariance | 96 | 8.883560 | 501 | 0.131488 | 0.144307 |
| ETTh2 | iTransformer | Rolling correlation | 96 | 3.733014 | 1251 | 0.132000 | 0.137118 |
| ETTm1 | Autoformer | Inverted FARM | 96 | 6.675320 | 1251 | 0.054679 | 0.058591 |
| ETTm1 | Crossformer | Rolling covariance | 96 | 31.776340 | 1501 | 0.054328 | 0.079632 |
| ETTm1 | PatchTST | Mutual information | 96 | 0.618097 | 501 | 0.029224 | 0.029406 |
| ETTm1 | TimeXer | Inverted rolling correlation | 96 | 0.809044 | 751 | 0.028265 | 0.028496 |
| ETTm1 | iTransformer | Inverted smooth random walk | 96 | 4.226735 | 751 | 0.029112 | 0.030397 |
| ETTm2 | Autoformer | Inverted rolling covariance | 96 | 8.066759 | 751 | 0.112769 | 0.122664 |
| ETTm2 | Crossformer | Inverted rolling correlation | 96 | 56.020943 | 501 | 0.097762 | 0.222292 |
| ETTm2 | PatchTST | Rolling correlation | 96 | 1.063221 | 1001 | 0.063732 | 0.064417 |
| ETTm2 | TimeXer | Entropy | 96 | 2.546961 | 1251 | 0.066090 | 0.067818 |
| ETTm2 | iTransformer | Inverted rolling covariance | 96 | 10.076355 | 751 | 0.068177 | 0.075816 |
| TrafficL | Autoformer | FARM | 96 | 6.734518 | 751 | 0.234853 | 0.251811 |
| TrafficL | TimeXer | Inverted rolling correlation | 96 | 2.661655 | 501 | 0.163954 | 0.168437 |
| TrafficL | iTransformer | Inverted rolling correlation | 96 | 5.581958 | 501 | 0.147666 | 0.156396 |
| Weather | Autoformer | Rolling correlation | 96 | 34.942647 | 1001 | 0.005192 | 0.007980 |
| Weather | Crossformer | Inverted rolling covariance | 96 | 76.640501 | 1501 | 0.001418 | 0.006072 |
| Weather | PatchTST | Inverted rolling correlation | 96 | 20.056259 | 1501 | 0.001277 | 0.001598 |
| Weather | TimeXer | Inverted mutual information | 96 | 2.782328 | 501 | 0.001297 | 0.001334 |
| Weather | iTransformer | Random walk | 96 | 9.919953 | 501 | 0.001256 | 0.001394 |

In addition, to show more specifically how the morphing process affects predictions, we provide, in Figure 3, prediction comparisons between morphed input data and raw input data, which is called "identity" function morphing. Figure 3 follows the same constructor and window length combinations as in Table 5.

We notice most of the selected predictions in Figure 3 is closer to the ground truth when compared to the same offset prediction, the prediction behavior changes as well as the y axis offset, improving forecasting performance overall. Selected offsets for these predictions are specified in the results jupyter notebook of the repository.

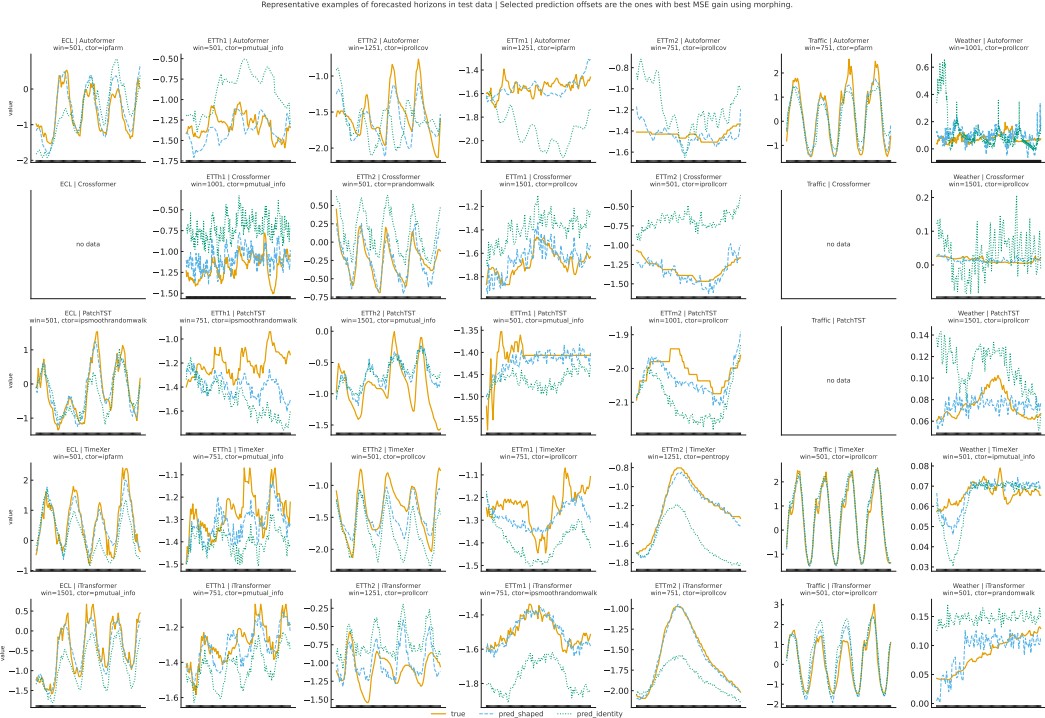

Figure 3: Representative examples of forecasted horizons in test data. Each subplot shows the ground truth series (solid line), predictions with morphed exogenous variables (pred_shaped, dashed line), and predictions with raw exogenous variables (pred_identity, dotted line). Subplots are arranged with datasets as columns and models as rows, x axis represents time but its values are removed for the sake of saving space, y axis is the actual time series value; each panel indicates the corresponding window length and morphing function that yielded the best overall mean squared error (MSE) gain in test for that dataset–model pair, and show the graph corresponding to the best MSE gain prediction offset for the 96 steps horizon. This visualization highlights the impact of exogenous data morphing across a diverse set of benchmarks and forecasting architectures.

## C  SIGNIFICANCE TESTS

**Scope and baselines.**    We evaluate whether morphing functions improve forecasting accuracy under a conservative design. Analyses are restricted to prediction length 96 (full ablation). Baselines are *identity* (raw input, i.e., no morphing) and *pure random walk*; *smooth random walk* is excluded for clarity.

**Blocks and normalization.**    Let a *block* be a dataset–model pair. For each block, let $E_{\mathrm{id}}$ denote identity MSE. For a morphing configuration $m$ (function and window), let $E(m)$ be its MSE, and define the relative improvement

$$r(m) = 1 - \frac{E(m)}{E_{\mathrm{id}}} \, .$$

We summarize morphing within each block in two complementary ways: (i) **Typical morphing** $E_{\mathrm{typ}} = \mathrm{median}_m E(m)$ (median across all morphing functions and windows), (ii) **Tuned morphing** $E_{\mathrm{tun}} = \min_m E(m)$ (best function+window). Random walk is window–independent; we use $E_{\mathrm{rw}}$ and $r_{\mathrm{rw}} = 1 - E_{\mathrm{rw}}/E_{\mathrm{id}}$.

**Motivation for inverted morphing functions.**    The morph ratio function is not always clear to measure "useful information". For example: for correlation-based morphing, it is not obvious whether "more correlation" is always better:

- High correlation means target and exogenous are very similar — but may add little new information.

- Low correlation may mean irrelevance — but can also bring complementary signals.

To account for this ambiguity, we extrapolated this thought to all morphing functions and also tested **inverted morphing functions**, where the morphing ratio is defined as $1 - \text{minmax}(\text{correlation}(t))$ instead of $\text{minmax}(\text{correlation}(t))$. This allows both strong and weak correlation regimes to be explored.

**Motivation for random walk baselines.** To ensure that improvements from morphing are not obtained merely by chance, we include **pure random walk** morphings. These inject exogenous signals that are stochastic and non-informative. If morphing functions consistently outperform random walk, it indicates that gains come from exploiting genuine statistical structure in the exogenous series.

## C.1 GLOBAL TESTS: TYPICAL VS TUNED

For each dataset–model block we obtain a distribution of morphing errors relative to identity. Because error distributions are often non-normal, heavy-tailed, and contain outliers, we use the **Wilcoxon signed-rank test** (a non-parametric paired test) instead of a $t$-test. This test only relies on the relative ordering of improvements and is robust to outliers, making it suitable for our setting. We also summarize morphing effects with the **median** across morphing configurations, since the median is more robust than the mean when a few extreme runs deteriorate accuracy. Thus, reported numbers should be interpreted as "typical" rather than "average" behavior.

Across the 32 dataset–model blocks we find:

- **Typical vs identity:** median improvement $\tilde{r}_{\text{typ}} = -0.2\%$; Wilcoxon signed-rank $p = 0.405$ (not significant). Interpretation: in the typical case, morphing is statistically indistinguishable from raw inputs.

- **Typical vs random walk:** median improvement $\tilde{r}_{\text{typ}} = -0.2\%$ vs $\tilde{r}_{\text{rw}} = -0.5\%$; Wilcoxon $p = 0.53$ (not significant). Interpretation: typical morphing does not beat a random baseline.

- **Tuned vs identity:** median improvement $\tilde{r}_{\text{tun}} = +6.1\%$; Wilcoxon $p = 1.17 \times 10^{-6}$ (significant). Interpretation: when tuned, morphing reduces error by about $6\%$ relative to identity on median.

- **Tuned vs random walk:** Wilcoxon $p = 2.56 \times 10^{-8}$ (significant). Interpretation: tuned morphing consistently outperforms random exogenous noise.

In summary, **typical morphing** is neutral compared to both identity and random walk, while **tuned morphing** provides clear, significant improvements. This can also be seen in Figure 4, which shows a heatmap of improvements for each dataset-model pair so it is easier to highlight best and worst cases. This leads to a distinction that reinforces morphing is not universally better, but when the right morphing function and window are chosen, it can deliver substantial gains.

## C.2 MORPHING FUNCTION RANKING (WINDOW–AGNOSTIC)

To avoid window-size multiplicity, we compute within each dataset–model block the **median MSE per morphing function across all window lengths**. We then assign **ranks** within each block (1 = best, $N$ = worst), so that functions are compared fairly without being advantaged by having more window candidates. Finally, we average these ranks across all blocks, producing a *global rank score*, shown in Table 6. Lower average ranks indicate more reliable morphing functions across datasets and models.

Entropy-based morphing functions lead on average, with FARM and correlation-based variants closely following.

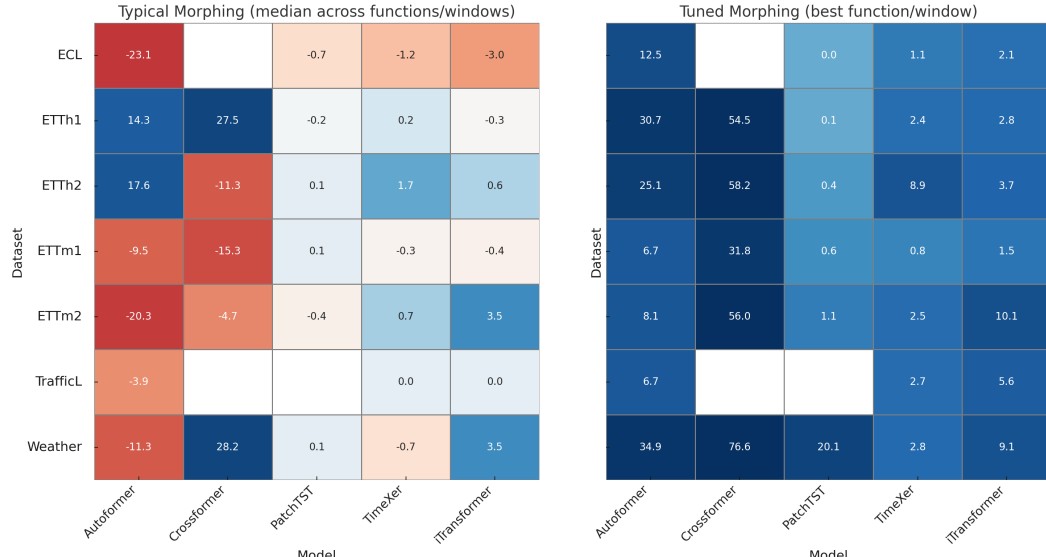

Figure 4: **Relative improvement vs identity** across datasets (rows) and models (columns). Left: *Typical morphing* (median across morphing functions and windows). Right: *Tuned morphing* (best morphing function and window). Blue $> 0$ indicates improvement, red $< 0$ deterioration; values in cells are % changes. White boxes mean there was not experiment in this combination, due to out of memory.

Table 6: Average rank of morphing functions across all datasets and models (median across windows within a block). Lower is better.

| Morphing function | Average rank |
|---|---|
| pentropy | 4.86 |
| ipentropy | 5.05 |
| ipfarm | 5.22 |
| iprollcorr | 5.25 |
| pfarm | 5.25 |
| pmutual_info | 5.47 |
| ipmutual_info | 5.72 |
| prollcov | 5.87 |
| iprollcov | 6.12 |
| prollcorr | 6.15 |

## C.3 WINDOW-LENGTH EFFECTS

We study the effect of window length on MSE using both an overall curve and per-dataset trends, which is shown in Figure 5. Short windows ($\approx 500$) tend to help industrial/abrupt datasets (e.g., ECL, ETTh2), while medium windows ($\approx 750$–$1000$) fit periodic/finer-grained datasets (ETTm, Weather, TrafficL). Very long windows ($\geq 1250$) generally dilute relevance.

## C.4 LOCAL WINS

Although global typical effects are neutral, tuned morphing shows *clear local wins* for several dataset–model pairs. Examples include:

- ETTh1–Crossformer and ETTh2–Autoformer/PatchTST: strong consistent improvements.
- **TimeXer**: gains in ETTh1, ETTh2, and ETTm2.

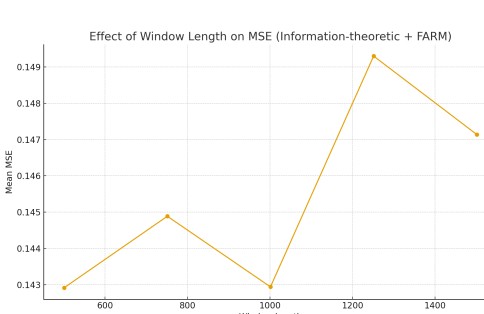
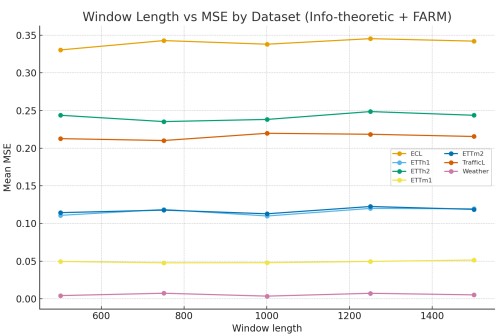

(a) Overall mean MSE vs window length (typical).   (b) Mean MSE vs window length by dataset (typical).

Figure 5: Effect of window length on forecasting accuracy (median across morphing functions).

- **iTransformer**: gains in ETTh2, ETTm2, and Weather.

These cases illustrate the potential of morphing when properly tuned.

### C.5    CONCLUSION

Morphing is *not* universally better when used blindly (typical median effect $\approx 0\%$). However, when the morphing function and window are chosen appropriately for a given dataset–model pair, it can *significantly* boost accuracy (tuned median gain $\approx 6\%$, $p \ll 0.001$). Future work should focus on predicting/selecting suitable morphing functions *a priori* (e.g., via dataset meta-features or light pilot scans), rather than applying morphing indiscriminately.

## D    FORMAL DEFINITION OF THE MORPHING FRAMEWORK

**Setup.**    We consider discrete time steps $t = 1, \ldots, T$.

- $y_{1:T}$: target time series, with scalar $y_t \in \mathbb{R}$.
- $x_{1:T}^{(k)}$: the $k$-th exogenous series (channel), $k = 1, \ldots, K$, with scalar $x_t^{(k)} \in \mathbb{R}$.
- $w \in \mathbb{N}$: morphing window length (context size).
- $S(\cdot; \theta)$: morphing function (statistical operator) with parameters $\theta$ (e.g., correlation, covariance, entropy, mutual information, FARM).
- $\mathcal{T} \subseteq \{w, \ldots, T\}$: calibration index set for normalization.
- $\varepsilon > 0$: small constant to avoid division by zero in normalization.
- Optional: $\ell \in \mathbb{N}$ smoothing length (if $\ell = 1$ there is no smoothing).

**Step 1: Rolling statistic (per channel).**    For each channel $k$ and time $t \geq w$, compute a windowed statistic

$$s_t^{(k)} \;=\; S\big(x_{t-w+1:t}^{(k)}, \, y_{t-w+1:t}; \, \theta\big).$$

Typical examples include:

$$S_{\mathrm{corr}}(x, y) = \rho(x, y) \quad \text{(Pearson or Spearman)},$$
$$S_{\mathrm{cov}}(x, y) = \mathrm{cov}(x, y),$$
$$S_{\mathrm{H}}(x) = H(x) \quad \text{(entropy via histogram/KDE)},$$
$$S_{\mathrm{MI}}(x, y) = I(x; y) \quad \text{(mutual information)},$$
$$S_{\mathrm{FARM}}(x, y) = f\big(|\rho(x, y)|\big) \quad \text{(correlation-derived)}.$$

Important: the morphing framework is not limited to statistical morphing functions, any function that takes two time series and outputs a single time series may be used.

**Step 2: Normalization to a morphing ratio.** Normalize $s_t^{(k)}$ to a ratio in $[0, 1]$ using min–max scaling over $\mathcal{T}$:

$$\tilde{s}_t^{(k)} \;=\; \mathrm{clip}\!\left(\frac{s_t^{(k)} - \min_{u \in \mathcal{T}} s_u^{(k)}}{\max_{u \in \mathcal{T}} s_u^{(k)} - \min_{u \in \mathcal{T}} s_u^{(k)} + \varepsilon},\; 0, 1\right).$$

Because it is a priori unclear whether "more correlation" is always better, we allow an inverted variant:

$$r_t^{(k)} \;=\; \begin{cases} \tilde{s}_t^{(k)}, & \text{standard morphing,} \\[4pt] 1 - \tilde{s}_t^{(k)}, & \text{inverted morphing.} \end{cases}$$

(Optional) Temporal smoothing of $r_t^{(k)}$ over the past $\ell$ steps:

$$\bar{r}_t^{(k)} \;=\; \frac{1}{\ell} \sum_{i=0}^{\ell-1} r_{t-i}^{(k)}, \qquad \text{and we reassign } r_t^{(k)} \leftarrow \bar{r}_t^{(k)} \text{ if } \ell > 1.$$

**Step 3: Build morphed inputs through gating.** Morphing *gates* each exogenous value by its ratio:

$$z_t^{(k)} \;=\; r_t^{(k)} \cdot x_t^{(k)}, \qquad \mathbf{z}_t \;=\; \big(z_t^{(1)}, \ldots, z_t^{(K)}\big).$$

The forecasting model $f_\phi$ (e.g., Autoformer, PatchTST, TimeXer, iTransformer) consumes the morphed inputs $\mathbf{z}_{1:T}$ (and the target history if applicable). No augmentation is used; the original exogenous values are *replaced* by their gated versions.

**Compact operator form.** Define the morphing operator $\mathcal{M}_{S,w}$ acting on $(x^{(k)}, y)$ by

$$\mathcal{M}_{S,w}\big(x^{(k)}, y\big)_t \;=\; r_t^{(k)} \cdot x_t^{(k)}, \quad \text{with} \quad r_t^{(k)} \;=\; \mathcal{I}\Big(\mathcal{N}\big(S(x_{t-w+1:t}^{(k)}, y_{t-w+1:t})\big)\Big),$$

where $\mathcal{N}$ is the min–max normalization on $\mathcal{T}$ and $\mathcal{I}$ optionally applies inversion ($u \mapsto 1 - u$) and smoothing.

**Notation summary.**

- $y_{1:T}$: target series; subscript $1:T$ denotes indices $1$ through $T$.
- $x_{1:T}^{(k)}$: $k$-th exogenous series; superscript $(k)$ indexes the channel.
- $t$: time index; $w$: window length; $K$: number of exogenous channels; $T$: sequence length.
- $S(\cdot; \theta)$: morphing statistic with parameters $\theta$; examples in Step 1.
- $s_t^{(k)}$: raw rolling statistic; $\tilde{s}_t^{(k)}$: normalized statistic in $[0, 1]$.
- $r_t^{(k)}$: morphing ratio (possibly inverted and/or smoothed).
- $z_t^{(k)}$: gated exogenous value used as model input.
- $\mathcal{T}$: calibration index set for min–max scaling; $\varepsilon$: small constant for numerical stability.
- $\ell$: smoothing length (if $\ell = 1$ there is no smoothing).

**Pseudo-code.**

```
Inputs:
  y[1:T]              # target series
  X[1:T, 1:K]         # exogenous channels
  W                   # window length (int)
  STAT                # morphing function: corr, cov, entropy, MI, FARM, ...
  invert              # boolean flag for inverted morphing
  smooth_len          # smoothing length (1 = no smoothing)

Procedure:
  # 1) Rolling statistic per channel
```

```
for k = 1..K:
  for t = W..T:
    xw = X[t-W+1 : t, k]
    yw = y[t-W+1 : t]
    s[k,t] = STAT(xw, yw)

# 2) Min-max normalization to [0,1] on calibration set Tcal
for k = 1..K:
  s_min = min_{t in Tcal} s[k,t]
  s_max = max_{t in Tcal} s[k,t]
  for t = W..T:
    r = (s[k,t] - s_min) / (s_max - s_min + eps)
    if invert: r = 1 - r
    r = clip(r, 0, 1)
    R[k,t] = r

# 3) Optional smoothing
if smooth_len > 1:
  for k = 1..K:
    R[k, W..T] = moving_average(R[k, W..T], window = smooth_len)

# 4) Gating: replace exogenous with gated values
for k = 1..K:
  for t = 1..T:
    Z[t,k] = R[k,t] * X[t,k]

Return Z   # morphed (gated) exogenous inputs
```

# E   COMPARISON TO THE CATS FRAMEWORK

**Overview of CATS.**   CATS (*Constructing Auxiliary Time Series*) builds **auxiliary time series (ATS)** Lu et al. (2024) from the original multivariate inputs (OTS) and uses them as exogenous variables to capture inter–series relations. A predictor (which can be simple, e.g., an MLP) jointly forecasts OTS and ATS, after which a **linear projection** maps the joint prediction back to the OTS space and is added as a residual correction to the OTS forecast. In practice, this yields an effect *akin* to 2D temporal–contextual attention while remaining architecture–agnostic (no modification to attention layers).

**Similarities.**

- **Cross variable information focus** Our morphing framework and CATS both aim to **leverage relationships among input variables** to improve forecasting accuracy.
- **Generate new relevant information from raw inputs** Both recognize that raw exogenous series often contain signals that are hard for models to understand how it relates to the related phenomenon of interest, and that some filtering or weighting is necessary.
- **Model dependence.** Both are model–agnostic. CATS does not modify attention; it augments the input space and uses a projection head. Morphing only changes inputs (gating) and leaves the predictor untouched.

**Differences.**

- **Input cardinality** CATS increases the number of input variables in the model while Morph keeps the same cardinality, thus ensuring the search space does not increase.
- **Flexibility.** CATS is coupled into the model during training while Morph can be run as a data preprocessing step.
- **Where information is injected.** CATS *adds* ATS channels and later *projects* their predictions back as a residual; our morphing *gates* each exogenous channel directly at the input

via a time–varying ratio $r_t^{(k)} \in [0, 1]$ computed from rolling statistics (correlation, entropy, MI, FARM, and inverted variants).

- **How relationships are captured.** CATS *learns* ATS from OTS and relies on the predictor + projection to funnel ATS information into OTS forecasts. Morphing *computes* windowed statistics between each exogenous and the target to selectively amplify/attenuate inputs before the model.

- **Focus** CATS originally tackles multivariate forecasting problem while Morph tackles "forecast with exogenous" problem.

**Illustrative comparison.** Figure 6 highlights the difference in integration strategy: CATS constructs correlation-based auxiliary signals (*ATS*) that interact with the attention mechanism, while morphing applies transformations to exogenous series at the input stage.

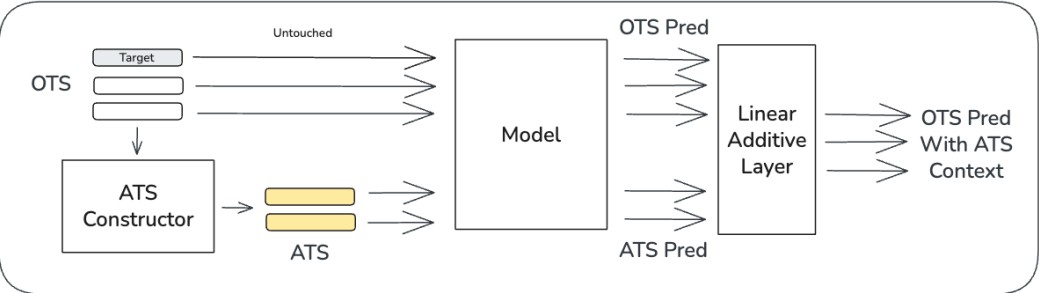

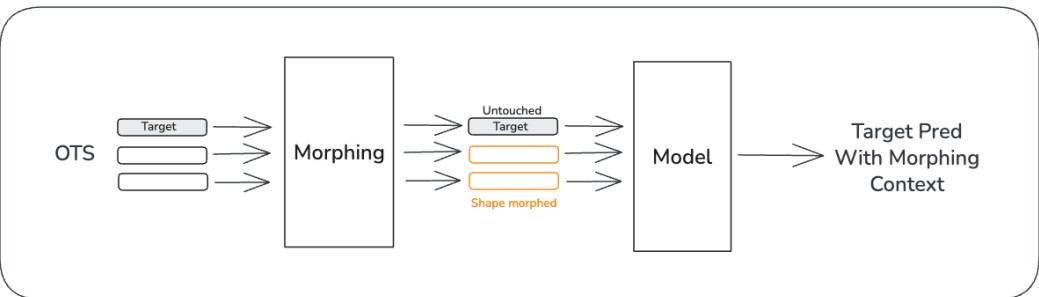

Figure 6: Comparison of the CATS framework (top) and the morphing framework (bottom). CATS modifies the model's attention layer with correlation-based auxiliary series, whereas morphing preprocesses exogenous inputs with morphing functions, leaving the model architecture unchanged.

**Summary.** Both approaches share the intuition that exogenous variables must be filtered or weighted according to their relationship among themselves in CATS and with respect to the target in Morph.

## F    MORPHING VS DLINEAR BASELINE

**Motivation.** DLinear is a strong channel–independent baseline. Since it processes each variable independently, morphing (which modifies exogenous signals to enhance cross–series interactions) has no effect on DLinear: its accuracy remains unchanged whether morphing is applied or not. In contrast, Transformers are channel-dependent models and consequently leverage cross–series interactions, and thus can benefit from morphing. The following experiments compare each Transformer (Autoformer, PatchTST, TimeXer, iTransformer, Crossformer) to DLinear at horizon 96.

**Table description.** Table 7 and Table 8 *Typical* $\Delta\%$ and *Tuned* $\Delta\%$ are the percentage MSE changes relative to raw inputs for the median morphing and for the best morphing, respectively. Positive values indicate lower error. *Overcame* = whether or not the Transformer was worse than DLinear with raw inputs but surpassed it with morphing. *Lost Adv.* (Lost advantage) = Whether a Transformer was better than DLinear with raw inputs but fell behind after typicalmedian morphing. *Typ* $\Delta\%$ *(mean/med)* and *Tun* $\Delta\%$ *(mean/med)* are the mean/median percentage MSE changes relative to the DLinear model inputs:

$$\Delta\% = \frac{\text{MSE}_{\text{DLinear}} - \text{MSE}_{\text{morph}}}{\text{MSE}_{\text{DLinear}}} \times 100.$$

Empty cells mean "no"; $\sqrt{}$ means "yes". Last row is a "TOTAL" row, which represents average percentage improvement in MSE or share of cases.

Table 7: Tuned Morphing vs DLinear at horizon 96: $\Delta\%$ relative to DLinear MSE. Positive = Transformer better than DLinear.

| Dataset | Model | Raw $\Delta\%$ | Tuned $\Delta\%$ | Overcame | Lost Adv. | Even better |
|---|---|---|---|---|---|---|
| ECL | Autoformer | 4.27 | 16.25 | | | $\sqrt{}$ |
| ECL | PatchTST | 19.23 | 19.23 | | | |
| ECL | TimeXer | 30.59 | 31.34 | | | $\sqrt{}$ |
| ECL | iTransformer | 28.02 | 29.53 | | | $\sqrt{}$ |
| ETTh1 | Autoformer | -80.51 | -25.18 | | | |
| ETTh1 | PatchTST | 11.94 | 11.99 | | | $\sqrt{}$ |
| ETTh1 | TimeXer | 12.14 | 14.24 | | | $\sqrt{}$ |
| ETTh1 | iTransformer | 8.97 | 11.53 | | | $\sqrt{}$ |
| ETTh1 | Crossformer | -475.67 | -162.04 | | | |
| ETTh2 | Autoformer | -46.04 | -9.44 | | | |
| ETTh2 | PatchTST | -1.22 | -0.85 | | | |
| ETTh2 | TimeXer | -7.13 | 2.39 | $\sqrt{}$ | | |
| ETTh2 | iTransformer | -1.79 | 2.01 | $\sqrt{}$ | | |
| ETTh2 | Crossformer | -294.92 | -65.01 | | | |
| ETTm1 | Autoformer | -65.63 | -54.57 | | | |
| ETTm1 | PatchTST | 16.87 | 17.39 | | | $\sqrt{}$ |
| ETTm1 | TimeXer | 19.45 | 20.10 | | | $\sqrt{}$ |
| ETTm1 | iTransformer | 14.07 | 15.35 | | | $\sqrt{}$ |
| ETTm1 | Crossformer | -125.11 | -53.58 | | | |
| ETTm2 | Autoformer | -70.31 | -56.57 | | | |
| ETTm2 | PatchTST | 10.56 | 11.51 | | | $\sqrt{}$ |
| ETTm2 | TimeXer | 5.84 | 8.24 | | | $\sqrt{}$ |
| ETTm2 | iTransformer | -5.26 | 5.34 | $\sqrt{}$ | | |
| ETTm2 | Crossformer | -208.63 | -35.73 | | | |
| TrafficL | Autoformer | 28.01 | 32.86 | | | $\sqrt{}$ |
| TrafficL | TimeXer | 51.85 | 53.13 | | | $\sqrt{}$ |
| TrafficL | iTransformer | 55.29 | 57.78 | | | $\sqrt{}$ |
| Weather | Autoformer | -41.30 | 8.07 | $\sqrt{}$ | | |
| Weather | PatchTST | 71.71 | 77.38 | | | $\sqrt{}$ |
| Weather | TimeXer | 76.37 | 77.03 | | | $\sqrt{}$ |
| Weather | iTransformer | 75.31 | 77.57 | | | $\sqrt{}$ |
| Weather | Crossformer | -7.51 | 74.89 | $\sqrt{}$ | | |
| TOTAL | ALL | -27.83 | 6.63 | 16% | 0% | 53% |

**Interpretation.** Typical morphing shows mixed improvements: some models deteriorate slightly relative to raw inputs when compared to DLinear. Tuned morphing, however, yields substantial and consistent error reductions, with many positive *Tun* $\Delta\%$. The "Overcame" column in Table 7 confirms that in $16\%$ of cases, tuned morphing made Transformers become better than DLinear, and in $53\%$ of the cases, morphing makes the model even better. While typical morphing shows a total of $19\%$ of either further improvement or overcoming. Tuned morphing does not make any model lose advantage over DLinear and typical morphing only made this happen in $3\%$ of the cases.

**Conclusion.** These results reinforce that **morphing helps to close the gap between channel–independent linear models (DLinear) and channel–dependent Transformers**. With tuning,

Table 8: Typical Morphing vs DLinear at horizon 96: $\Delta\%$ relative to DLinear MSE. Positive = Transformer better than DLinear.

| Dataset | Model | Raw $\Delta\%$ | Typical $\Delta\%$ | Overcame | Lost Adv. | Even better |
|---------|-------|------|---------|----------|-----------|-------------|
| ECL | Autoformer | 4.27 | -17.87 | | √ | |
| ECL | PatchTST | 19.23 | 18.62 | | | |
| ECL | TimeXer | 30.59 | 29.76 | | | |
| ECL | iTransformer | 28.02 | 25.89 | | | |
| ETTh1 | Autoformer | -80.51 | -54.67 | | | |
| ETTh1 | PatchTST | 11.94 | 11.80 | | | |
| ETTh1 | TimeXer | 12.14 | 12.36 | | | √ |
| ETTh1 | iTransformer | 8.97 | 8.71 | | | |
| ETTh1 | Crossformer | -475.67 | -317.28 | | | |
| ETTh2 | Autoformer | -46.04 | -20.34 | | | |
| ETTh2 | PatchTST | -1.22 | -1.16 | | | |
| ETTh2 | TimeXer | -7.13 | -5.33 | | | |
| ETTh2 | iTransformer | -1.79 | -1.18 | | | |
| ETTh2 | Crossformer | -294.92 | -339.42 | | | |
| ETTm1 | Autoformer | -65.63 | -81.35 | | | |
| ETTm1 | PatchTST | 16.87 | 16.92 | | | √ |
| ETTm1 | TimeXer | 19.45 | 19.17 | | | |
| ETTm1 | iTransformer | 14.07 | 13.75 | | | |
| ETTm1 | Crossformer | -125.11 | -159.48 | | | |
| ETTm2 | Autoformer | -70.31 | -104.89 | | | |
| ETTm2 | PatchTST | 10.56 | 10.20 | | | |
| ETTm2 | TimeXer | 5.84 | 6.50 | | | √ |
| ETTm2 | iTransformer | -5.26 | -1.60 | | | |
| ETTm2 | Crossformer | -208.63 | -223.01 | | | |
| TrafficL | Autoformer | 28.01 | 25.22 | | | |
| TrafficL | TimeXer | 51.85 | 51.85 | | | |
| TrafficL | iTransformer | 55.29 | 55.29 | | | |
| Weather | Autoformer | -41.30 | -57.26 | | | |
| Weather | PatchTST | 71.71 | 71.72 | | | √ |
| Weather | TimeXer | 76.37 | 76.22 | | | |
| Weather | iTransformer | 75.31 | 76.18 | | | √ |
| Weather | Crossformer | -7.51 | 22.85 | √ | | |
| TOTAL | ALL | -27.83 | -25.99 | 3% | 3% | 16% |

morphing not only reduces error systematically but often enables Transformers to overtake DLinear, while DLinear cannot leverage morphing because of its channel-independent architecture.

