# OpenReview forum: "Time Series Forecasting: Empowering Exogenous Data with Shape Morphing"
_ICLR.cc/2026/Conference — Submitted to ICLR 2026_

### Official Review · Reviewer_ayV8 · 2025-10-26

**Soundness:** 2
**Presentation:** 1
**Contribution:** 2
**Rating:** 2
**Confidence:** 4

**Summary:**

This paper proposes a morphing framework that adaptively reshapes exogenous time series before forecasting. Specifically, for each channel and time step, a morphing function computes a ratio from the local relationship between the exogenous input and the target series and amplifies useful intervals accordingly. Results show that morphing is a simple yet effective way to enhance the utility of exogenous information.

**Strengths:**

- The paper is well-structured and easy to follow.

- The proposed morphing approach builds on a preceding statistical analysis to identify the temporal influence of the exogenous series on the target variable, which is simple yet effective.

**Weaknesses:**

- The overall novelty is limited. A key component of the morphing framework is temporal saliency detection, yet there is no additional design or novel contribution specifically addressing this module.

- The experimental implementation details are missing. The authors select five transformer-based models, which exhibit substantial architectural diversity; in particular, Autoformer maps the channel dimension directly to the hidden dimension. Consequently, it remains unclear how the proposed morphing framework is adapted or applied to each architecture. A detailed explanation of how the framework is instantiated for each model is required.

- The experimental results in Table 2 are unclear. It lacks explanations of why experiments conducted on the same datasets, with the same models and saliency detection settings, yield different results. In addition, terms such as ipfarm and pentropy require clarification.

**Questions:**

- There is a significant performance decline for the Autoformer architecture on the Weather dataset. Could the authors provide a more in-depth analysis of this issue?

---

> ### Author Response · Authors · 2025-11-19
>
> Dear reviewer,
>
> Thank you for your review and your concerns. Please find your comments following.
>
> ---
>
> # Weaknesses:
>
> **1**: The paper discusses a novel method for integrating exogenous information in time series modelling. In doing so, it demonstrates the potential on an extensive ablation test including state-of-the-art deep-learning modelling
> approaches. The novelty lies in the concept for providing exogenous information with temporal salience related attention to the modelling with significantly lower complexity. Thus, the paper focuses on the potential of the
> proposed low complex, statistical approach to improve forecast accuracy and excludes research on temporal saliency detection. The results demonstrate substantial improvements for the proposed concept compared to the
> moderate accuracy gains yielded with increased complexity in most recent state-of-the-art deep-learning methods.
>
> **2**: The shape morphing method is model agnostic. Thus the framework remained the same (i.e. is a set of window length and saliency detection methods) for all applied architectures. The paper presents the
> potential of a new concept towards a new method. Thus, the experiment comprises a large ablation test for drawing best possible conclusions. Yet, we will review
> this section and discuss the experiment in more detail for more clarity.
>
> **3**: Thank you for this input. Indeed, this table misses the information about the different window sizes that result to different accuracy. We will also review the discussion of the table 2 and clarify the terminology used (pfarm -> FARM, prollcorr -> Rolling correlation, prollcov -> Rolling covariance, pmutual_info -> Mutual Information, pentropy -> Entropy). The "i" stands for inverted as discussed in the main text on line 385.
>
> # Questions:
>
> The ablation test for multiple different window sizes and morphing function was ran for the prediction horizon of 96 timesteps, as said in the main text.
> The -116% and -101% are related to prediction lengths of 192 and 336 because the window size and morphing function were not chosen for these horizons,
> we decided to include these results for the sake of comparison. Moreover, wondering about this behavior, the Autoformer uses auto-correlation and seasonal trends
> which are often naturally present in weather data (see weather data set extract https://anonymous.4open.science/r/AnonymousDropIn-547B/weather_dataset_subseries.jpg). So morphing these data may blur the natural auto-correlation in a way that the natural trend cycles worsen results depending
> on the forecasting horizon. Interestingly, we see an enhancement of 20% in mse for PatchTST (non-channel mixing model) on the same weather data for prediction length 96.
> Indeed, this is surprising but may be explained by the natural trends of other exogenous features of the weather dataset “showing” cycles regarding the target feature,
> which boosts overall patch/token mixing over time through the learning of the model (see weather data set extract https://anonymous.4open.science/r/AnonymousDropIn-547B/weather_dataset_subseries.jpg). This accuracy boost provided by morphing is enabling the non channel-mixing model (PatchTST) to "see" weather data channel-mixed features. This shows a very desired effect.
> The extract clearly shows the cyclic behaviour in variables as well as the strong positive and negative correlation between some exogenous variables and the target variable (OT).
>
> Please also take a look at the figure 3 and 4 in the appendix. These clearly show how morphing impacts predictions.

---

### Official Review · Reviewer_NcW4 · 2025-10-27

**Soundness:** 2
**Presentation:** 1
**Contribution:** 2
**Rating:** 2
**Confidence:** 4

**Summary:**

This paper proposes **Shape Morphing**, a model-agnostic preprocessing framework to solve the temporal saliency problem in time series forecasting.

The method computes a rolling statistical relationship (like correlation or mutual information) between the exogenous input and the target series, creating a dynamic morph ratio. This ratio is then used to gate the original input, amplifying its signal when it is relevant and attenuating it when it is not.

**Strengths:**

1. While many works have tried to incorporate exogenous variables by designing complex, model-specific attention mechanisms, this paper reframes the problem. It proposes Shape Morphing, a model-agnostic preprocessing framework. This model-external design is an original contribution that makes the solution highly generalizable. It can be applied to any forecasting model (new or existing) that accepts exogenous inputs, significantly broadening its potential impact and making it easy for practitioners to adopt.

2. This paper conducts a large ablation test across seven diverse datasets and five different state-of-the-art Transformer architectures (Autoformer, Crossformer, etc.). This rigorous validation demonstrates that the improvements are not an isolated phenomenon but are achievable across various model and data types, lending strong credibility to the method's effectiveness.

**Weaknesses:**

**1. Clarity and Paper Structure: The paper's overall clarity is a primary concern.**

- Unbalanced Focus: The writing in sections like Related Work (Section 2), as well as the experimental setup and conclusion, is overly verbose. These sections are excessively detailed, diminishing the focus on the paper's algorithm, which, by contrast, is not explained sufficiently.

- Core Algorithm in Appendix: The detailed algorithm of the Shape Morphing is explained in the appendix. This forces the reader to constantly switch between the main text and the appendix to build a complete understanding of the proposed method, which hinders readability.

- Lack of Intuitive Figures: This clarity issue is compounded by a lack of helpful visualizations. While Figure 2 provides a high-level overview, it fails to clearly illustrate how morphing impacts the final prediction. The paper would be significantly improved by a dedicated diagram illustrating the core algorithmic steps.

**2. Questions on Performance Significance: The empirical results raise questions about the method's true necessity and utility for modern architectures.**

- Inconsistent Gains: The paper highlights the large performance boost on Crossformer, but this result does not sufficiently represent the algorithm's universal utility. For other SOTA models that already perform well (e.g., PatchTST, TimeXer), the performance gains are minor and, in some cases, performance even degrades.

- Necessity for Modern Architectures: This inconsistency leaves a critical question unanswered: Is the Shape Morphing algorithm a truly necessary component for modern, robust SOTA architectures, or is its primary benefit in addressing the limitations of specific models?

**Questions:**

Please refer to weaknesses above

---

> ### Author Response · Authors · 2025-11-19
>
> Dear reviewer,
>
> Thank you for your review and your concerns. Please find your comments following.
>
> ---
>
> # Weaknesses:
>
> **1.1**: For better clarity of the contribution, we will extend the explanation of the new approach with more details. In addition we review the presentation according to the inputs of reviewer 36H3. Can you give examples of overly verbose sentences used in the paragraphs of any
> of these referred sections, please? Based on the examples, we will think of alternative writing styles
> or even removing parts.
>
> **1.2**: We intentionally put the details of the algorithm in the appendix,
> but we made sure to describe  the main part of the algorithm in the main text.
> Can you please point out what details present in the appendix would enhance the
> presentation of our work?
>
> **1.3**: Illustration of the toy-case in Figure 2 item d shows the comparison between prediction with and without morphing. Although both lines have overlaps, it clearly shows regions where they differ. Given that, can you elaborate more on how this figure fails do illustrate the morphing impacts? Additionally,
> we provided several comparisons for the real datasets in the appendix (see appendix B, Figure 3), showing how morphing impacts prediction with different dataset and model combinations at their best results. Regarding the suggestion of a diagram to describe the algorithmic steps, do you think of a flowchart similar to the one in the appendix section E (Figure 6) or the pseudo-code in the appendix D?
>
> **2.1**: Comparing the shape morphing gains to the original performance gains provided by the most recent SOTA models,  including PatchTST, TimeXer, and iTransformer (in TimeXer paper, the gains from the second best model are around 1% to 5% in MSE),
> the proposed novel concept yielded unprecedented improvements at its best combinations of morphing function and window size. This holds for 4 out of the 5 models: the PatchTST is the only one that doesn't profit most of the time because its architecture is non-channel mixing. The weather data set is the only one where the PatchTST profits substantially since this data set contains strong covariate variables and the best morphing function is covariance/correlation based. Indeed, the concept is still on an early stage it presents significant gains compared to improvements with increased complexity in inner models (PatchTST, TimeXer).
> Thus, it is worth to continue researching on the proposed approach.
>
> **2.2**: These "inconsistent" but coherent gains are explained by the transformer models characteristics: Autoformer and Crossformer have the heavy channel mixing architectures based on cross attention or multivariate attention, thus they both profit a lot from morphing.
> TimeXer and iTransformer have light channel mixing architecture that use different techniques, and they profit consistently from morphing, but not as much as Crossformer and Autoformer.
> Lastly, PatchTST almost does not change its performance due to having no cross channel architectural design. This model treats subseries as tokens (patches) and only works with token mixing and not with channel mixing.
> Thus results are inconsistent because each transformer model architecture behaves differently, and our work coherently shows the strength of shape morphing when considering cross channel information. Notice that shape morphing improves the SOTA MSE accuracy of the TimeXer by up to 8.9% (see appendix Figure 4).

---

> > ### Comment · Reviewer_NcW4 · 2025-11-27
> >
> > Thank you for your detailed response. After careful consideration, I have decided to maintain my score. Here is a summary of my remaining concerns:
> >
> > **1. Paper Structure & Content Balance**
> >
> > While I appreciate the authors' willingness to revise, the main text needs to be self-contained. Even if the full derivation is lengthy, the core logic and mechanism of the algorithm must be clearly presented in the main paper, with only specific technical details reserved for the Appendix. Currently, it feels like the essential explanation has been offloaded to the Appendix, which forces the reader to rely on supplementary material to fully grasp the proposed method.
> >
> > **2. Generalizability**
> >
> > The clarification regarding PatchTST confirms that the method provides limited benefit (or even degradation) for current SOTA channel-independent architectures. While the improvements on other models are noted, this limitation suggests the method is not as universally applicable to modern baselines as claimed.

---

### Official Review · Reviewer_36H3 · 2025-10-28

**Soundness:** 1
**Presentation:** 1
**Contribution:** 2
**Rating:** 2
**Confidence:** 4

**Summary:**

The paper investigates the problem of forecasting with exogenous variables, where multiple auxiliary time series are used to predict a single target variable. The authors highlight the phenomenon of temporal saliency of exogenous variables, referring to the varying relevance of external inputs over time. To better exploit this property, the paper proposes a morphing framework that adaptively reshapes exogenous time series before forecasting. For each channel and time step, a morphing function computes a ratio from the local relationship between the exogenous input and the target series, amplifying informative intervals accordingly. Experiments conducted on multiple long-horizon forecasting benchmarks show that morphing can yield notable improvements for certain model–dataset combinations, suggesting that this approach effectively enhances the utility of exogenous information.

**Strengths:**

1.The proposed framework refines how exogenous signals contribute to target prediction, introducing time-dependent adaptive processing that adds interpretability and practical value.
2.The approach is conceptually simple, easy to integrate with existing time series method, and potentially useful in real-world multivariate forecasting scenarios.

**Weaknesses:**

1.The experimental evaluation is relatively limited, covering only five deep learning models in the main results. Moreover, while the paper focuses primarily on a data-processing perspective, it lacks broader comparisons with classical statistical, machine learning, and time-series pretraining frameworks. Expanding the experiments to include these baselines would provide a more convincing assessment of the proposed framework’s generality, effectiveness, and position within the broader landscape of time-series research.

2.The selection of datasets is rather narrow. Including additional benchmarks such as EPF, which is widely used in studies involving exogenous variables, would significantly enhance the credibility and comprehensiveness of the results.

3.The improvements reported in Table 1 are not consistent across all experimental settings. In particular, models like TimeXer and PatchTST exhibit little or no gain, which raises concerns about the robustness and reliability of the proposed morphing mechanism.

4.The paper suffers from several presentation and structural issues. The Related Work section is missing, and parts of the literature review are scattered within the Method section, which obscures the connection between the proposed approach and prior studies. The overall organization of the paper is unclear, with weak transitions between sections. Moreover, the dataset descriptions in the Experiment section could be moved to the Appendix for better readability, and the Conclusion section is overly lengthy and repetitive, diluting the main findings. A clearer structure and more concise presentation would substantially improve the paper’s readability and impact.

**Questions:**

Since the morphing operation locally rescales the exogenous series in the numerical domain, could this distort the original temporal patterns of exogenous variables and negatively impact the tokenization or representation learning of these covariates?

---

> ### Author Response · Authors · 2025-11-19
>
> Dear reviewer,
>
> Thank you for your review and your concerns. Please find your comments following.
>
> ---
>
> # Weaknesses:
>
> **1**: Primarily, the focus of this work is to showcase the effect when morphing exogenous time series for modelling. It reveals that statistical analysis with low complexity and morphing input series can
> significantly support models in increasing forecast accuracy. We believe that the results based on five deep learning models (which are representative of the channel-mixing and non channel-mixing
> transformer architectures  - other channel-independent models do not change output of the target depending on the exogenous, so it does not make sense to include channel independent models in the ablation)
> provide enough evidence to continue researching on this effect. Can you suggest other models to consider? Again, this work presents a novel idea for including exogenous data in modelling with low complexity.
> Thus, it does not present a final approach that requires a full-fledged evaluation. Can you suggest what other pretraining frameworks we should consider?
>
> **2**: Primarily we focused on long-term forecasting and the EPF data set is typically used for STF. Thus we explicitly excluded this data set from our analysis.
>
> **3**: This paper presents a new method how to include exogenous information for modelling. The goal was not to present a novel approach that consistently
> increases accuracy throughout all experiment setups. Instead, the main find of this paper is to show that there is way to increase accuracy better than
> inner model architectural modifications in a model-agnostic way. Finding the best way to come up with a morphing function and window size for each
>  dataset and model combination is worth a future work. Moreover, the gains yielded with morphing are substantial compared to the increase with models
> like TimeXer.
>
> **4**: Other reviewers consider the paper as "well structured and easy to follow". Despite different naming, the paper presents a "Related Work" and a "Method" section. Still, we consider renaming
> sections and reviewing the structure to further increase the readability of the paper.
>
> # Questions:
>
> True. Morphing distorts temporal patterns aiming to benefit forecasting of the target. The distortion only negatively impacts the forecast when weakly tuned with incorrect window size and morphing function (see appendix Figure 4, typical morphing: this shows median behavior across different window size and morphing functions without optimal tuning).

---

### Official Review · Reviewer_exuv · 2025-10-30

**Soundness:** 2
**Presentation:** 3
**Contribution:** 3
**Rating:** 4
**Confidence:** 4

**Summary:**

The paper proposes a simple preprocessing framework that rescales each exogenous channel over time using a morph ratio derived from rolling, lag-aware statistics (e.g., correlation, covariance, entropy, mutual information) computed between that channel and the target series. The goal is to emphasize intervals where the exogenous signal is predictive (“temporal saliency”) and attenuate it elsewhere, before feeding inputs to forecasting models. The approach targets the observation that channel-dependent Transformers often fail to capitalize on exogenous features. The paper reports broad ablations on 7 long-horizon benchmarks and 5 Transformer families.

**Strengths:**

- The motivation is significant, where exogenous variables are typically informative only in specific intervals and irrelevant elsewhere.
- The idea is simple and model-agonistic.
- Temporal saliency detection is an interesting experiment to me.

**Weaknesses:**

- In the empirical analysis, it seems the method only brings a substantial gain on Crossformer, but just marginal on more recent models, such as PatchTST, TimeXr, and iTransformer.
- Many baselines use “original hyperparameters” (no tuning), while morphing gets extensive tuning. This risks an uneven playing field.
- Statements that Transformers are “permutation-invariant” and “cannot learn channel dependency” are somewhat over-broad; many architectures inject positional encodings and learn cross-channel structure.
- For ECL and Traffic, results are incomplete due to “exhausted computational power,” weakening the generality of claims precisely on the hardest multivariate settings (hundreds of exogenous channels).

**Questions:**

- Can you report the average gain for different models?
- Were the experiments run many times for a fair comparison?
- Was morph function + window size selected on validation only?

---

> ### Author Response · Authors · 2025-11-19
>
> Dear reviewer,
>
> Thank you for your review and your concerns. Please find your comments following.
>
> ---
>
>
> # Weakness:
>
> **issue 1**: this is not true. The TimeXer and iTransformer models also profit substantially from the morphing approach. Indeed not to the same extent as the crossformer or autoformer, but still substantially (around 1% to 10% jumps from best results of their original approach: see appendix Figure 4) when compared to the increase of accuracy provided by their recent inner architectural improvements (jumps around 1% to 5% from the second best model, as reported in their original papers).
>
> **issue 2**: the tuned hyperparameters of the inner model remains the same for the approach "without morphing" exogenous series as well as for the approach "with morphing". Thus,
> both approaches have seen the same inner model (transformer) hyperparameters. So, the comparison is fair. Secondly, the "tuned" scenario corresponds to the selection of the best experiment results out of the full ablation
> test. More precisely, the tuning implies the selection of the best combination of moving window size and morphing function (mutual information, correlation, e.t.c.).
> Typical and tuned scenarios are not applied on the baseline ("without morphing"). Instead, typical and tuned scenarios only refer to our "with morphing"-approach.
>
> **issue 3**: our statement is too strong. We will rephrase it to: The self-attention mechanism widely used in transformer models is permutation-invariant. We acknowledge that the
> permutation-invariance does not hold in general for transormer-models. Yet, their ability to learn cross-channel dependencies is weak has been surpassed by linear models as in the work of \cite{Zeng et al. 2023}
>
> **issue 4**: this indeed weakens, but not too much. We have 7 data sets and 5 models (in total 35 combinations). The missing combinations include 3 out of 35 for partially missing results and only 3 out of 35 without results at all.
> We think that, this does not hinder us from drawing conclusions about the effect of morphing. If additional results helps for increasing the rating of the paper, we will do our best to run the missing computational extensive
> experiments in a larger GPU.
>
> # Questions:
>
> **1**: Yes we can. We decided to use the median value to have comparable values that are less sensitive to large variations or outliers. If this helps to improve the rating, we will comment it here and add it to the paper.
>
> **2**: No. Refer to the main text and the appendix. See line 697.
>
> **3**: No. Both, the morphing function and the window size were kept as variables for the whole test. The results in the main text
> present combinations for the best scenario results and combinations for typical scenario results.
> The best results are selected (from the full ablation combinations) based on the test data set.
> The paper explains this in detail in the appendix on the lines 803-806.

---

> > ### Comment · Reviewer_exuv · 2025-11-25
> >
> > For my question 3, a common approach is to set different random seeds, for example {2021, 2022, 2023, 2024, 2025}, run the experiment multiple times, and then take the average result.
> >
> > Considering the overall experimental setup, I have decided to keep my existing score. Having said that, I still believe this paper is valuable and worthy of further improvement.

---

### Meta-Review · Area_Chair_tdeZ · 2026-01-07

**Summary:**

## Summary

The strengths of the work can be summarized as follows:

- The motivation of exploiting temporally salient exogenous variables is clear and practically relevant [exuv, 36H3].
- The Shape Morphing framework is simple, model-agnostic, and easy to integrate into existing models [NcW4, exuv, 36H3, ayV8].
- Broad ablation studies show that morphing can improve performance in certain model–dataset settings [NcW4].

The weaknesses of the work can be summarized as follows:

- The novelty is limited, as the approach mainly rescales inputs using known statistics [ayV8].
- Performance gains are inconsistent and mainly observed on specific architectures, with limited benefits on recent SOTA models [36H3, NcW4].
- The experimental setup raises concerns about fairness and robustness due to best-case selection of morphing settings [exuv, 36H3, ayV8].
- Some important results are missing due to computational constraints, weakening general conclusions [36H3, exuv].
- Presentation and clarity issues remain, with key details placed in the appendix and figures insufficiently illustrative [36H3, NcW4, ayV8].

**Reviewer Concerns:**

**Reviewer exuv**

- Addressed: The authors clarified that morphing also yields gains for TimeXer and iTransformer, and that tuning applies only to the morphing process, not the backbone models.
- Outstanding: The reviewer still questions the statistical significance and robustness of the reported gains, due to the lack of repeated runs with different random seeds.

**Reviewer 36H3**

- Addressed: The authors justified their focus on five Transformer models, explaining that other channel-independent models are not suitable for ablation, and clarified the exclusion of certain datasets.
- Outstanding: Concerns remain regarding the limited breadth of baselines, inconsistent improvements across models, and serious presentation issues, including missing or poorly organized related work, scattered dataset descriptions, and overly verbose sections that obscure the main contribution.

**Reviewer NcW4**

- Addressed: The authors provided additional explanations on why different architectures benefit differently from shape morphing and defended the inclusion of detailed algorithmic descriptions in the appendix.
- Outstanding: The reviewer remains concerned that the main paper is not self-contained. The core logic and mechanism of the algorithm must be clearly presented in the main text rather than relying on the appendix, and the method’s general applicability to modern, channel-independent architectures is limited.

**Reviewer ayV8**

- Addressed: The authors clarified terminology, explained anomalous results, and provided context on different prediction lengths.
- Outstanding: The reviewer still considers the novelty weak, finds experimental details insufficiently clear across different architectures, and questions whether the focus on temporal saliency detection constitutes a meaningful new contribution.

**Reviewer Scores:**

**Reviewer exuv**: Likely to maintain score 4.

This reviewer sees value in the idea, but remains cautious due to concerns about experimental fairness and robustness. The rebuttal clarifies some issues but does not fully resolve doubts about statistical significance and generality.

**Reviewer 36H3**: Likely to maintain score 2.

This reviewer expresses strong concerns about experimental scope, robustness, and paper organization. The rebuttal does not sufficiently address the perceived limitations in evaluation breadth and clarity.

**Reviewer NcW4**: Likely to maintain score 2.

Despite acknowledging the potential of a model-agnostic preprocessing approach, the reviewer remains unconvinced about the necessity and universal usefulness of Shape Morphing for modern SOTA architectures, and about the clarity and completeness of the main paper.

**Reviewer ayV8**: Likely to maintain score 2.

The reviewer continues to view the work as incremental, with experimental details across architectures insufficiently clear, and view the novelty as limited.

---

### Decision · Program_Chairs · 2026-01-26

Reject